# Possible Use of Blood Tryptophan Metabolites as Biomarkers for Coronary Heart Disease in Sudden Unexpected Death

**DOI:** 10.3390/metabo10010006

**Published:** 2019-12-19

**Authors:** Kobchai Santisukwongchote, Yutti Amornlertwatana, Thanapat Sastraruji, Churdsak Jaikang

**Affiliations:** 1Department of Forensic Medicine, Faculty of Medicine, Chiang Mai University, Chiang Mai 50200, Thailand; kobchai.san@gmail.com; 2Center of Excellence in Oral and Maxillofacial Biology, Faculty of Dentistry, Chiang Mai University, Chiang Mai 50200, Thailand; s_thanapat@hotmail.com

**Keywords:** picolinic acid, tryptophan, tryptophan metabolites, coronary heart disease, sudden unexpected death

## Abstract

Coronary heart disease (CHD) is the major cause of death in sudden unexpected death (SUD) cases. Tryptophan (TRP) and its metabolites are correlated with the CHD patient but less studies in the SUD. The aim of this study was to evaluate the relationship of TRP and its metabolites with the CHD in the SUD cases. Blood samples and heart tissues were collected from CHD subjects (*n* = 31) and the control group (*n* = 72). Levels of kynurenine (KYN), kynurenic acid (KYA), xanthurenic acid (XAN), 3-hydroxyanthranillic acid (HAA), quinolinic acid (QA), picolinic acid (PA) and 5-hydroxyindoleacetic acid (HIAA) were determined by HPLC-DAD. A severity of heart occlusion was categorized into four groups, and the relationship was measured with the TRP metabolites. The HIAA and The KYN levels significantly differed (*p* < 0.01) between the CHD group and the control group. Lower levels of QA/XAN, PA/KA, HAA/XAN, KYN/XAN and KYN/TRP were found in the CHD group. However, PA/HAA, PA/HIAA, PA/KYN and XAN/KA values in the CHD group were higher than the control group (*p* < 0.05). This study revealed that the values of PA/KA and PA/HAA provided better choices for a CHD biomarker in postmortem bodies.

## 1. Introduction

Sudden unexpected death (SUD) is defined as a natural, nonviolent, unexpected death occurring within twenty-four hours of the onset of symptoms [1]. A SUD is found in about 56% of medicolegal cases in Thailand [2]. Cardiovascular disease (CVD), including heart and blood vessel disorders, hypertension, coronary heart disease (CHD), cerebrovascular disease, peripheral vascular disease, heart failure, rheumatic heart disease, congenital heart disease and cardiomyopathies, is the major cause of morbidity and mortality worldwide [3]. CHD is the most common cause of SUD [4].

Under the Thai criminal justice system, causes of death in SUD cases are necessary to be identified. Many cases lacked reliable medical documents regarding underlying diseases that could help to explain the causes of death. Both external examination of the heart gross morphology and microscopic findings were appropriate methods for SUD diagnosis.

Cutting the surface of the coronary artery, a type of heart gross morphology examination, is an important procedure for CHD diagnosis [5]. Sometimes, the CHD diagnosis by using this process was not an accurate method and depended on randomized cutting at lesion area. Therefore, the microscopic method helps to confirm the existing CHD. However, in some cases, an autopsy cannot be performed due to different cultural and spiritual believes. In this situation, a cardiac biomarker is an alternative solution for a CHD diagnosis.

Cardiac troponin level in serum has become an increasingly important biomarker for myocardial injury in many cardiovascular diseases, especially in patients with CHD [6,7], and in cardiovascular cases in the forensic field. However, the cardiac troponin might not be specific or suitable enough as a cardiac biomarker in postmortem samples [8].

The kynurenine (KYN) pathway plays an important role in cardio pathophysiology and catabolism pathway is shown in Figure 1. Many studies have indicated changes of KYN and tryptophan (TRP) were found in CHD patients [9,10,11]. There is an increase of inflammatory cytokines when atherosclerosis occurs, which can induce indoleamine 2,3-dioxygenase (IDO) enzyme, a rate-limiting enzyme in TRP catabolism [12]. The IDO is highly up-regulated by immune activation and inflammation and appears to be important in the pathogenesis of CHD [13]. KYN/TRP value reflects the IDO activity found in CHD patients [14] but in SUD has been less studied.

The aim of this study was to determine the downstream of TRP metabolites, both the KYN and serotonin pathways, in order to find out a relation with coronary artery disease in SUDs. The blood concentration levels of TRP and its metabolites including KYN, kynurenic acid (KA), 3-hydroxyanthranillic acid (HAA), xanthurenic acid (XAN), quinolinic acid (QA), picolinic acid (PA) and 5-hydroxyindoleacetic acid (HIAA) in SUDs diagnosed as CHD were compared with those of deaths from noncoronary artery diseases. Furthermore, we compared levels of the metabolites with degree of coronary occlusion.

## 2. Results

Levels of TRP and its metabolites were found in all subjects, and mean values are shown in Table 1. Mean levels of HIAA and KYN differed between the CHD and the control group. A high level of HIAA was found in the CHD group (*p* = 0.018) but the KYN level was higher than the control group (*p* < 0.001). Mean ratios of PA/HAA, PA/HIAA, PA/KYN and XAN/KA were significantly increased in the CHD group (*p*-values of 0.043, 0.028, 0.001 and 0.015, respectively). Meanwhile, the QA/XAN, PA/KA, HAA/XAN, KYN/XAN and KYN/TRP levels were significantly higher than those of the control group (*p*-values of 0.018, 0.044, 0.029, 0.001 and 0.002, respectively). The ratio of TRP and its metabolites reflected the enzyme activities in the TRP catabolism pathway; remarkably, KYN/TRP acting as IDO activity. There were no significant differences in mean levels of QA, PA, HAA, XAN, TRP, KA and other ratios between groups. In this study, the age of the subjects ranged 21–86 years. Mean age in the CHD group (58.16 ± 13.34 years) was similar to the control group (52.89 ± 13.90 years).

Pearson’s correlation was used for evaluating a relationship between the TRP metabolite levels and the grade of coronary occlusion. The QA, PA, HAA, KYN, TRP and KA levels and ratio values of the metabolites were significantly correlated with the degree of coronary occlusion; the results are shown in Table 2. The QA, HAA, KYN, TRP and KA levels showed negative correlations (*p* = 0.003, *p* = 0.001, *p* < 0.001, *p* < 0.001, *p* = 0.001, respectively). Only the PA level showed a positive correlation (*p* < 0.001). The ratios of PA/KYN, PA/KA and PA/HAA were more positively correlated than the other ratios, while the ratios of QA/PA and KYN were negatively correlated.

Box plots revealed the relationship between the grading of occlusion and statistically correlated metabolites (Figure 2). The values of the KYN and KYN/TRP were significantly decreased depending on the degree of coronary artery narrowing. Interestingly, the PA/HAA and PA/KA levels in grades 2, 3 and 4 of occlusion were significantly higher than those in grade 1.

Trends of the levels of TRP and its metabolites in terms of degree of coronary artery severity are summarized and shown in Table 3. The levels of XAN, PA and HIAA increased according to degree of coronary artery occlusion; also, the PA level was suitable for indicating occluded coronary artery. While the levels of TRP, KYN, KA, HAA and QA decreased, the TRP and HAA levels were less affected in the CHD group.

## 3. Discussion

In this work, TRP and its metabolites in blood samples of the 103 males who died from sudden death were investigated. There were different levels of serum TRP and KYN among males and females [14]. We selected only male subjects who had a postmortem interval of within 24 h. To decrease factors affecting TRP metabolism, the subjects who exhibited decomposition, did not have an accurate determination of the time of death or suffered from malignant disease, stroke and tuberculosis were excluded [15,16].

The quantified levels of KA, TRP, and HAA in the SUD sample showed similar ranges compared with normal human serum [17]. The levels of XAN, KYN, HIAA, PA and QA were higher than normal levels in human subjects [18,19,20]. Levels of TRP metabolites in normal human plasma significantly differed between ethnicities [17]. There have been a few studies about the TRP pathway in Asian ethnicities. Difference in dietary culture [21] and high-TRP diets can influence TRP metabolite levels. At the present, there are still limited data about postmortem blood levels of TRP and its metabolites in all the TRP pathways.

CHD induces oxidative stress in the human body, but physiological defense mechanisms boost antioxidant systems to limit oxidative stress. 2-amino−3-hydroxymuconic-6-semialdehyde is a substrate of both PA and QA [22]. Synthesis of PA, an antioxidant molecule, might affect QA synthesis by competing against the substrate.

Although the PA level was not significantly increased in the CHD group, it had a positive correlation with the occlusion grading. Accumulation of PA coexisted with the reduction of QA downstream catabolism in the severe CHD cases. In Figure 2, the values of PA/HAA and PA/KA in non-occluding coronary artery (grade 1) had the lowest level and significantly differed from the other grades. These ratios could segregate grade 1 from the other grades in the CHD.

Some observed values of 52 ratios reflected enzyme activities in the TRP catabolic pathway for example, KYN/TRP presented as IDO activity and KYN/KA acted as kynurenine amino transferase activity. Not only were the PA/HAA, PA/HIAA, PA/KYN increased in the CHD group but also the PA/HAA, PA/HIAA, PA/KYN, PA/XAN, PA/TRP and PA/KA had positive correlation with the degree of coronary artery occlusion. These findings suggested that PA production might be increased instead of QA production in CHD, leading to the QA/PA having negative correlation. Products ratios in KYN pathway and serotonin pathway were demonstrated in the PA/HIAA value. The positive correlation reflected higher coronary occlusion and might be shifted through the KYN pathway.

In Table 1, no evidence indicates that the QA level was significant among groups, but we found that the QA level correlated with the degree of coronary artery narrowing (Table 2). The decrease of the QA level reflected the severity of coronary occlusion resulting from the negative correlation of the QA/PA. The 2-amino-3-hydroxymuconic-6-semialdehyde might be shifted into the PA pathway in the CHD group. This process was an oxidative stress defense mechanism which decreased pro-oxidant QA level and increased antioxidant PA molecules [23,24]. Normally, the levels of QA and PA are equal [24]. QA plays a main role in neurological disorder, as it is a neuroactive metabolite of HAA and acts as an excitotoxin that is a high-potency agonist of N-methyl-D-aspartate receptors associated with hypertension, myocardial infarction and unstable atherosclerotic plaque [25]. Energetic deficits, behavioral alterations and other neurological diseases are toxic effects of the QA molecule [26].

HAA acts as an antioxidant and anti-inflammatory substance in the oxidative stress process [27]. HAA inhibits atherosclerosis by regulating lipid metabolism [28] and has a significant function in antioxidant systems and the anti-inflammatory process [27]. The decrease of the HAA level in the CHD group was similar to that in the control group, but in the CHD group had a negative correlation with the degree of coronary artery occlusion (*p* = 0.001). Another study showed that the HAA acted as an pro-oxidant since it was catabolized into QA [29]. We found that a decrease of the HAA level in severely occluded coronary artery state occurred to limit the cellular damage response from QA [30]. A protective mechanism might occur via PA synthesis or 2-amino-3-hydroxymuconic-6-semialdehyde production. These phenomena had positive correlations with PA/HAA value.

HIAA is a final product of serotonin pathway. The HIAA level was significantly higher in the CHD group than in the control group. There was no statistically significant correlation between the HIAA level and grading of coronary artery occlusion. The HIAA/KYN value had a positive correlation, which indicated no fluxing through serotonin pathway for the CHD group. The KYN might be catabolized due to the HIAA level not showing statistically significant correlation with the degree of coronary occlusion, and the KYN level was lower than its metabolites level in the CHD group. The HIAA did not correlate with coronary occlusion but it had a significant difference with the CHD group, because pneumonia and sepsis were suggested as causes of death in the control group. Infection state might decrease plasma serotonin, the substrate of HIAA, by inhibiting serotonin transporter in platelets [31].

The XAN level was similar among groups, and no correlation was found with degree of vessel occlusion. Both negative correlation of the QA/XAN and KYN/XAN and positive correlation of the XAN/KA indicated the increase in the XA pathway within the CHD group. The PA/XAN value had a positive correlation, indicating that enzyme activity in XA pathway might be less than in the PA pathway. Under inflammatory condition, kynureninase enzyme was more dominant than KAT enzyme in CHD [32]. However, the XA has remained an obscure pathway in biological systems [33]. A synthesis of XAN is an important process to prevent 3-hydroxykynurenine accumulation, as this is probably a toxic substance [34]. The XAN acted as a potential antioxidant and had an iron-chelating property [35]. However, a study found that the XAN was pro-oxidant and might induce cytotoxicity [36].

KA is a neuroprotective substance and acts as an antagonist to the NMDA receptor [27]. Dysregulation of endogenous KA might cause of hypertension, myocardial infarction and maternal hypertension [37,38]. The KA also acted as a reducing agent and scavenged hydroxyl radicals [39]. The KA levels were not difference between groups, similar to the results of Zuo et al. [33]. The KA level correlated with the degree of coronary occlusion and the ratios of PA/KA, XAN/KA and TRP/KA had a positive correlation. These results suggested that the KA pathway was decreased in severity of vessel narrowing.

The TRP level negatively correlated with the state of occlusion. The TRP level in the CHD group was decreased [26,33,40]. The ratios of QA/TRP and HAA/TRP showed a negative correlation but the TRP/KA was positively correlated. These findings indicated that production rate of the QA, HAA and KA were less than the TRP catabolism rate in the CHD group. The PA/TRP had a positive correlation from increasing flux through PA pathway.

The values of KYN and KYN/TRP in the CHD group were lower than in the control group. The KYN value in the grade 1 occlusion was significantly different with the values in grades 3 and 4. The KYN/TRP significantly differed between grades 1 and 4. The KYN and the KYN/XAN ratio had a negative correlation, while the PA/KYN had a positive correlation. These results demonstrated that decrease of the KYN level depended on the severity of coronary narrowing. Enzymes in the KYN pathway might have lower activity than those in the PA and XAN pathways. Our results revealed that the TRP metabolites in postmortem differed from the CHD patients. The IDO activity is a key enzyme in the TRP metabolism which could be reduced in hypoxia condition and declined in KYN production [41]. Postmortem changes might affect TRP and its metabolites, however this needs to be studied further.

The control group in Table 1 was composed of the occlusion in grades 1 to 3, and grade 4 was only for the CHD group. Thus, mean levels of QA, PA, HAA, TRP and KA in the grades 1 to 3 showed no significant differences compared with the grade 4, but the metabolites still had a correlation with the degree of occlusion.

Many noncommunicable diseases, including type 2 diabetes, nonalcoholic fatty liver disease and obesity, affect the IDO activity in TRP metabolic pathway [10]. Patients with higher TRP level tended to present higher levels of insulin resistance, triglycerides and blood pressure [42]. IDO was up-regulated in older patients and related to increased susceptibility of aged liver to nonalcoholic fatty liver disease development [43]. Therefore, the autopsy cases with diabetes, hypertension and hyperlipidemia should be interpreted as CHD with TRP and its metabolites carefully.

In conclusion, the PA/KA and the PA/HAA were the most suitable for classifying non-CHD out of the CHD in SUDs. The decrease of the QA, HAA, KA, KYN, TRP levels and increase of the PA and XAN levels reflected the degree of occlusion. This is the first study about a prospectively evaluated coronary artery occlusion using the TRP metabolic pathways in postmortem and confirmed by coronary occlusion pathology. The PA/KA and PA/HAA could be used for excluding non-CHD from CHD in SUDs under more than 75% occlusion, graded as level 4 condition. However, our study had several limitations. Firstly, small sample size might cause data misinterpretation. Secondly, the study group was limited to specific geographic area which might not reflect a generalized area. Finally, postmortem changes and other factors might affect TRP and its metabolites.

## 4. Materials and Methods

### 4.1. Subjects and Study Design

One hundred three males who died from sudden death and underwent autopsy at Maharaj Nakorn Chiang Mai Hospital, Department of Forensic Medicine, Faculty of Medicine, Chiang Mai University were selected in this study. The subjects who had malignant disease, stroke, tuberculosis disease, age under 15 years and decomposition were excluded. We also excluded subjects who had any evidence of underlying disease such as diabetes, hypertension and hyperlipidemia. Postmortem interval range was specified to be less than 24 h. Thirty-one subjects who were diagnosed with severe occluded coronary artery (occlusion more than 75% of the cut surface, or the cause of death being the CHD group) were included in the study group. The flowchart of the case selection is shown in Figure 3. Written informed consent was obtained from direct relatives. The study protocol was approved by the Research Ethics Committee Faculty of Medicine, Chiang Mai University (FOR-2561-05497).

### 4.2. Collection and Specimen Preparation

Femoral blood was collected (about 5 mL) in a sodium fluoride tube and stored at −20 °C before analysis. Three milliliters of the blood samples were mixed with acetonitrile (2 mL). The solution was shaken for 5 min and centrifuged at 5500 rpm for 5 min; then, the supernatant was collected. The residues were re-extracted with 2 mL acetonitrile two times. The supernatants were combined and evaporated under nitrogen gas. The residues were reconstituted with 20 mM sodium acetate buffer before analysis with HPLC-DAD.

### 4.3. Measurement of Tryptophan and Its Metabolites

The levels of TRP, KYN, KA, XA, HAA, QA, PA and HIAA in plasma were measured by high-performance liquid chromatography with diode array detector (HPLC-DAD) by modified method of Cseh [44]. The sample analysis was performed using an Agilent LC 1260 infinity binary pump system. Gradient elution of two solvents composing a 20 mM sodium acetate buffer pH 6.4 adjusted with acetic acid (A) and acetonitrile (B) was used for detection. The total run time of the program was 16 min. Gradient elution program was begun with 100% of solvent A and was held at this concentration for 0–6 min. This was followed by 95% of solvent A for 6–7 min and then reduced to 80% of solvent A for the next 7–14 min and then increased to 100% of solvent A for the next 14–16 min with flow rate was 0.7 mL/min. An HPLC Chromolith^®^ Performance RP-18e analytical column (100 × 4.6 mm) was used for stationary phase. The spectrums were determined by diode array at 220 to 400 nm for identification or TRP and its metabolites.

### 4.4. Pathology

Heart tissues were collected and prepared under routine processes of the Department of Pathology, Faculty of Medicine, Chiang Mai University. The heart samples were stained with hematoxylin and eosin (H&E). Percentage of occlusion in heart tissues was calculated by OLYMPUS Cell Sens Standard 2.2 program. Degree of coronary artery occlusion was separated into four grades following percent occlusion: grade 1, occluded less than 25%; grade 2, occluded 25–49%; grade 3, occluded 50–74%; and grade 4, greater than 75%. The grade 1, 2, 3 and 4 samples were from 51, 9, 14 and 29 subjects, respectively. The narrowing could be concentric or eccentric, according to Yang et al.’s method, which indicated that severe narrowing of coronary with less than 25% patency in an area could lead to SUD [13].

### 4.5. Statistical Analysis

Descriptive statistics were expressed as mean ± S.D. All continuous variables were checked for normal distribution using Shapiro–Wilk normality test. Differences in blood TRP and TRP metabolite levels between the control group and the CHD group were compared using the Mann–Whitney U-test. Correlations between the TRP metabolite levels and degree of coronary occlusion were calculated using Pearson’s correlation coefficient. A *p-*value of less than 0.05 was considered significant.

## Figures and Tables

**Figure 1 metabolites-10-00006-f001:**
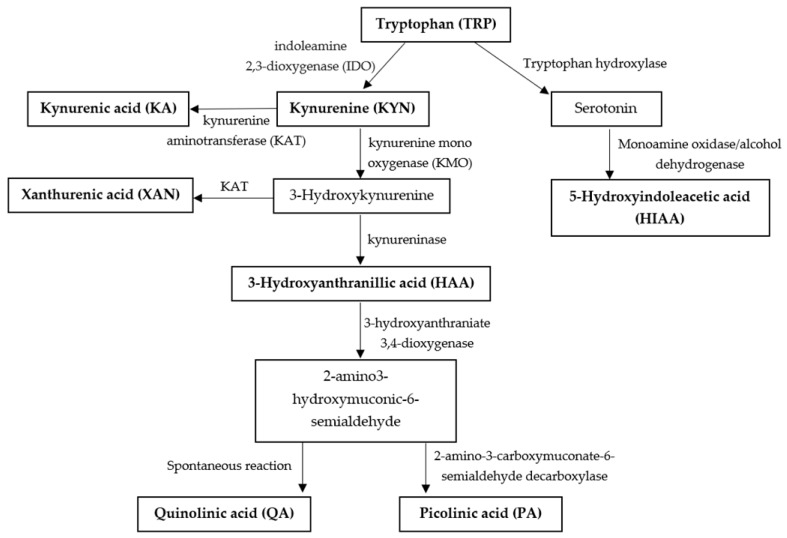
Schematic illustration of tryptophan (TRP) catabolism along with the kynurenine (KYN) and serotonin pathway.

**Figure 2 metabolites-10-00006-f002:**
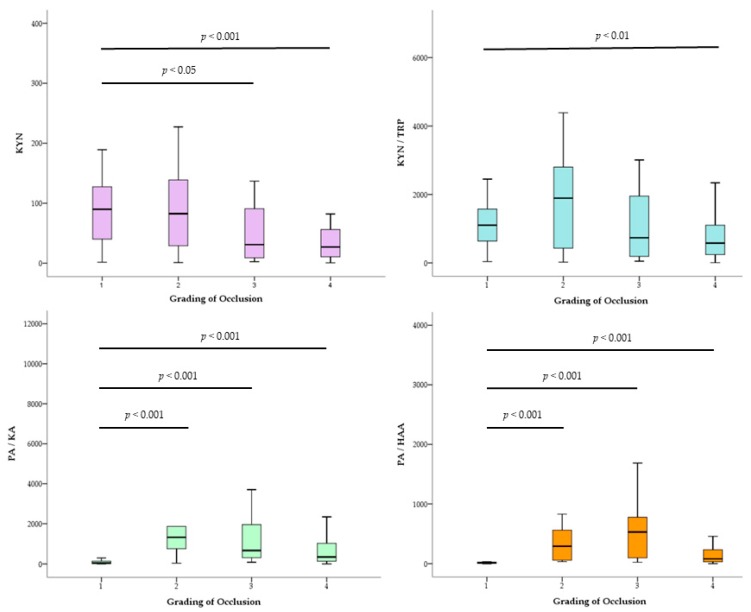
Comparison between blood concentration levels of KYN, KYN/TRP, PA/KA and PA/HAA ratios in each coronary occlusion graded by Mann–Whitney U-test. Statistical differences between the grades are indicated by *p*-value less than 0.05.

**Figure 3 metabolites-10-00006-f003:**
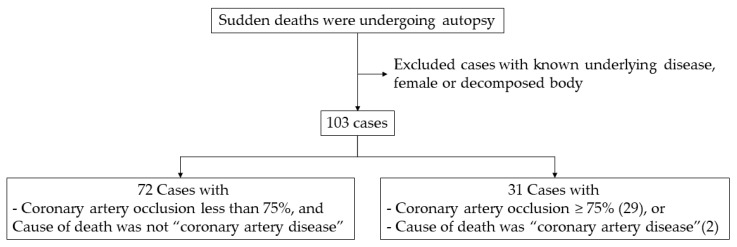
Flowchart of case selection in this study.

**Table 1 metabolites-10-00006-t001:** Blood concentration level of TRP and its metabolites, alongside some significant ratios, compared with the coronary heart disease (CHD) and the control groups.

Metabolites	Range	Total (*n* = 103)	CHD Group (*n* = 31)	Control Group (*n* = 72)	*p*-Value
TRP (mM)	0.04–0.17	0.06 ± 0.03	0.06 ± 0.02	0.07 ± 0.03	0.076
KYN (mM)	0.70–306.87	71.68 ± 60.88	41.98 ± 47.68	84.47 ± 61.74	<0.001
KA (mM)	0–0.07	0.005 ± 0.01	0.003 ± 0.004	0.007 ± 0.01	0.078
XAN (mM)	0–0.28	0.03 ± 0.04	0.03 ± 0.04	0.02 ± 0.04	0.131
HAA (mM)	0–0.88	0.02 ± 0.09	0.01 ± 0.03	0.02 ± 0.10	0.062
QA (mM)	0.04–4.07	0.61 ± 0.85	0.30 ± 0.23	0.74 ± 0.98	0.124
PA (mM)	0–7.81	0.74 ± 1.11	0.83 ± 1.04	0.70 ± 1.15	0.165
HIAA (mM)	0–364.59	22.75 ± 44.12	27.45 ± 70.41	20.73 ± 26.22	0.018
KYN/XAN (×10^4^)	0–17.83	1.48 ± 2.66	0.73 ± 1.51	1.78 ± 2.97	0.001
KYN/TRP (×10^3^)	0–7.00	1.15 ± 1.08	0.72 ± 0.70	1.34 ± 1.16	0.002
XAN/KA	0–229.84	18.83 ± 36.76	32.22 ± 53.56	13.06 ± 24.85	0.015
HAA/XAN	0–32.33	1.94 ± 3.94	1.17 ± 1.92	2.27 ± 4.52	0.029
QA/XAN (×10^3^)	0–1.30	0.11 ± 0.23	0.05 ± 0.07	0.14 ± 0.26	0.018
PA/HAA (×10^3^)	0–3.33	0.25 ± 0.52	0.33 ± 0.68	0.22 ± 0.44	0.043
PA/HIAA	0–4.37	0.13 ± 0.46	0.29 ± 0.81	0.06 ± 0.14	0.028
PA/KYN	0–0.45	0.03 ± 0.06	0.05 ± 0.09	0.02 ± 0.04	0.001
PA/KA (×10^4^)	0–1.11	0.07 ± 0.13	0.06 ± 0.07	0.07 ± 0.15	0.044
Age (years)	21–86	54.48 ± 13.88	58.16 ± 13.34	52.89 ± 13.90	0.056

The values are presented as means ± SD. A nonparametric Mann–Whitney U-test used for comparing between groups at *p* < 0.05. Abbreviations: QA = quinolinic acid; PA = picolinic acid; HAA = 3-hydroxyanthranillic acid; HIAA = 5-hydroxyindoleacetic acid; KYN = kynurenine; XAN = xanthurenic acid; TRP = tryptophan; KA = kynurenic acid.

**Table 2 metabolites-10-00006-t002:** Pearson’s correlation of TRP and TRP metabolite levels with degree of coronary occlusion.

Metabolites	Correlation Coefficient (*r*)	*p-*Value
TRP	−0.356	<0.001
KYN	−0.358	<0.001
KA	−0.322	0.001
HAA	−0.309	0.001
QA	−0.293	0.003
PA	0.361	<0.001
TRP/KA	0.215	0.029
KYN/XAN	−0.275	0.005
KYN/TRP	−0.253	0.01
XAN/KA	0.251	0.01
HAA/TRP	−0.228	0.02
HAA/XAN	−0.207	0.036
QA/PA	−0.425	<0.001
QA/XAN	−0.202	0.04
QA/TRP	−0.20	0.043
PA/HAA	0.437	<0.001
PA/HIAA	0.383	<0.001
PA/KYN	0.533	<0.001
PA/XAN	0.247	0.012
PA/TRP	0.403	<0.001
PA/KA	0.45	<0.001
HIAA/KYN	0.278	0.004

Statistical significance was determined using a nonparametric Mann-Whitney test at *p* < 0.05.

**Table 3 metabolites-10-00006-t003:** The trend summary report of TRP, TRP metabolites and significant ratios in the correlation with degree of coronary artery occlusion.

Metabolites	Trend Correlation with Coronary Occlusion Degree
TRP	−
KYN	−
KA	−
XAN	+
HAA	−
QA	−
PA	+
HIAA	+
KYN/TRP	−
PA/KA	+
PA/HAA	+

Results are presented as: (+), indicates the increasing of the metabolite level in the degree of coronary artery occlusion; (−), indicates the decreasing of the metabolite level in the degree of coronary artery occlusion.

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
