# Peer review of "Possible Use of Blood Tryptophan Metabolites as Biomarkers for Coronary Heart Disease in Sudden Unexpected Death"

_metabolites, 2019, doi:10.3390/metabo10010006_

Round 1
Reviewer 1 Report
In this manuscript, Santisukwongchote and colleagues aimed at identifying prognostic biomarkers of coronary heart disease in sudden unexpected death, by measuring tryptophan and its metabolites in post-mortem blood samples collected during 31 autopsies from subjects who died for coronary artery occlusion and from 72 control subjects who died for other causes of death with no lesions of coronary artery occlusion. Heart tissues were also collected and analysed for determining the degree of arterial occlusion by a histopathological score. Heart tissues were categorized into four groups according to the severity of occlusion. Correlations among the metabolite levels and the degree of occlusion were investigated and discussed. In particular the authors found that the ratios PA/KA and PA/HAA were significantly different between the CHD lesion and the other causes of death in the SUD and they suggest the use of these ratios as CHD biomarkers for post-mortem samples.
The manuscript is very difficult to read because it is not correctly written in English. Moreover typing errors are present throughout the text, capital letters appear randomly in the text, acronyms are introduced and not used …
The main concern of the reviewer relies on the experimental design and the conclusions that the authors drew from their results. How where the two classes of individuals selected? In particular, how where the 31 subjects belonging to the CHD SUD group selected? Was the diagnosis performed looking at the degree of arterial occlusion in heart tissues after death? This is very probable, so what is this research adding to the forensic scenario? This study describes an altered metabolic status, which is solely linked to the degree of arterial occlusion; it cannot be correlated to the cause of death. The altered metabolites can be considered as biomarkers of damage, and the same pattern would most probably be found in living individuals. So why performing blood HPLC analysis if histopathology is already highly informative on the severity of the disease? The set of described metabolites definitely has no prognostic value, and cannot be used to ascertain that the SUD is related to CHD, being the only information referred to the atherosclerotic involvement of the coronary tree as stated by the authors in the introduction.
A minor concern regards the exact number of CHD SUD cases. The authors firstly stated that they collected samples from 31 autopsies of individuals who died for coronary artery occlusion, but then they stratified the severity of arterial occlusion into four groups and considered as CHD SUD only 29 subject with degree of occlusion higher than 75%.
Globally the manuscript does not add any information exploitable in medical-legal cases.
Author Response
We are grateful to you for the time and constructive comments on our manuscript. We have implemented your comments and suggestions and wish to submit a revised version of the manuscript for further consideration in the journal. Below, we also provide a point-by-point response explaining how we have implemented reviewers’ comments. We look forward to the outcome of your assessment.
Comment 1: The manuscript is very difficult to read because it is not correctly written in English. Moreover typing errors are present throughout the text, capital letters appear randomly in the text, acronyms are introduced and not used…..
Answer: We apologize for our English language. We have revised and clarified by native speakers. Capital letters and other errors were check carefully.
Comment 2: How where the two classes of individuals selected?
Answer: In this study, we classified two groups by using percent of occlusion (coronary occlusion more than 75% for the CHD group and less than 75% for the control group).
Comment 3: In particular, how where the 31 subjects belonging to the CHD SUD group selected?
Answer: For study design; we calculated population size was 31 subjects. We included the 31 subjects from 103 cases and flowchart of case selection is shown in Figure 3.
Comment 4: Was the diagnosis performed looking at the degree of arterial occlusion in heart tissues after death?
Answer: The CHD diagnosis was performed by looking at the degree of arterial occlusion and confirmed by pathohistological method. All cases were diagnosed by forensic experts.
Comment 5: This is very probable, so what is this research adding to the forensic scenario?
Answer: This research study in forensic field.
Comment 6: This study describes an altered metabolic status, which is solely linked to the degree of arterial occlusion; it cannot be correlated to the cause of death. The altered metabolites can be considered as biomarkers of damage, and the same pattern would most probably be found in living individuals.
Answer: During living individuals, the metabolic status correlated to lesion the disease. We hypothesized that the metabolic status still correlated to the lesion after death. Our results showed that tryptophan and its metabolites in blood correlated with the CHD lesion which the same as living individuals.
Comment 7: So why performing blood HPLC analysis if histopathology is already highly informative on the severity of the disease?
Answer: Thanks for your suggestion. Although the histopathological method is highly informative, in some autopsy cases cannot be performed due to different local confession and culture of dead relatives. Under this situation, cardiac biomarker is an alternative solution for the CHD diagnosis. This sentences added in the manuscript, see page 2, line 44-45.
Comment 8: The set of described metabolites definitely has no prognostic value, and cannot be used to ascertain that the SUD is related to CHD, being the only information referred to the atherosclerotic involvement of the coronary tree as stated by the authors in the introduction.
Answer: Thank you for your suggestion. Indeed in this study, we presented about the relation of altered metabolic status with degree of vessel occlusion and the relation of tryptophan metabolites with CHD. Prognostic value will be present in further study.
Comment 9: A minor concern regards the exact number of CHD SUD cases. The authors firstly stated that they collected samples from 31 autopsies of individuals who died for coronary artery occlusion, but then they stratified the severity of arterial occlusion into four groups and considered as CHD SUD only 29 subjects with degree of occlusion higher than 75%.
Answer: The31 cases included to the CHD group; 29 cases had more than 75% coronary artery occlusion which diagnosed by pathology methods and 2 cases were diagnosed by forensic pathologist (percent occlusion nearly 75%). We added the details in Figure 3. Pages 9
Comment 10: Globally the manuscript does not add any information exploitable in medical-legal cases
Answer: Thank you for your comment. In this study, all data were proceeded under inform consents and the ethic committee. Provided data cannot use to identify to the subjects. Moreover, we did not expose any personal information. So, we assure that this study is right under medicolegal scope.
Reviewer 2 Report
The authors describe their work on the relationship of tryptophan and its metabolites on the CHD in sudden unexpected death (SUD) cases. It was found that tryptophan metabolism may be related with CHD in postmortem. Moreover, we found that the value of PA/KA and PA/HAA may be a good biomarker for CHD. This is an interesting study. Appropriate methodology has been employed and the conclusions appear to be justified based on the data at hand. Although the authors have indicated some limitations of the study. I have a few recommendations for consideration.
Introduction. Please provide a stronger rationale for the study as well as a clear hypothesis to be tested in the study. Methods. Analysis only conducted in males who died from SUD. No females used in this study? Results/Discussion. It would be helpful to the reader in order to understand the significance of tryptophan and metabolites in cardiovascular health, if the authors could provide a scheme depicting the mechanisms/target areas of tryptophan and metabolites. Discussion. I am not sure if the data are suggestive of a prognosis may be more diagnostic? Please comment. Discussion. Is there a possibility that altered tryptophan metabolism is predictive of higher risk of CHD in SUD? Discussion. Is there a possibility of sex differences in tryptophan metabolism and SUD? Please comment.Author Response
We are grateful to you for the time and constructive comments on our manuscript. We have implemented your comments and suggestions and wish to submit a revised version of the manuscript for further consideration in the journal. We also provide a point-by-point response explaining how we have implemented reviewers’ comments. We look forward to the outcome of your assessment.
Comment 1: Introduction. Please provide a stronger rationale for the study as well as a clear hypothesis to be tested in the study
Answer: We added a paragraph in introduction part to provide stronger rationale about diagnosing CHD in SUD, and we added some phrases to clarify the hypothesis, see page 2, line 41-45.
Comment 2: Methods. Analysis only conducted in males who died from SUD. No females used in this study?
Answer: In this study, we included only male subject because tryptophan and kynurenine level are different male and female. In Northern Thailand, male who died from SUD are more than female about 15 times and our research under Ethic approval within one year then we selected only male case for study.
Comment 3: Results/Discussion. It would be helpful to the reader in order to understand the significance of tryptophan and metabolites in cardiovascular health, if the authors could provide a scheme depicting the mechanisms/target areas of tryptophan and metabolites.
Answer: Thank you for your suggestion. We added Table3. “The trend summary report of the TRP, the TRP metabolites and significant ratios in the correlation with degree of coronary artery occlusion” in Result part, see page 6.
Comment 4: Discussion. I am not sure if the data are suggestive of a prognosis may be more diagnostic? Please comment.
Answer: Thank you for your comment. Indeed in this study, we meant to research about the relation of altered metabolic status with degree of vessel occlusion and the relation of tryptophan and its metabolites with CHD. The results showed that PA/KA and PA/HAA were good to classify non-CHD from CHD, but it cannot use for prognose or diagnose CHD. Then, we decided to delete the word, “prognostic”, out from the title to make it less confusing to the readers.
Comment 5: Discussion. Is there a possibility that altered tryptophan metabolism is predictive of higher risk of CHD in SUD?
Answer: From the results, it cannot predict high risk of CHD in SUD, but might be predict the degree of occlusion which more than 25 percent by using PA/KA and PA/HAA values which useful for precaution the CHD incident in human.
Comment 6: Discussion. Is there a possibility of sex differences in tryptophan metabolism and SUD? Please comment.
Answer: There were differences in normal tryptophan and kynurenine level among male and female which the level in female is lower than male. In SUD cases have been less study about tryptophan and kynurenine level in both male and female.
Reviewer 3 Report
Authors should present their data as mean plus minus SD and not SEM because readers are interested in knowing the dispersion of value and not the precision of the mean due to paucity of observations at least in one group. Authors correctly stated as limitations the possible undetected co-morbidities such as T2DM, Metabolic Syndrome and NAFLD and thus they should lessen of importance their findings pointing out with much emphasis this aspect and not using few words.
Author Response
We are grateful to you for the time and constructive comments on our manuscript. We have implemented your comments and suggestions and wish to submit a revised version of the manuscript for further consideration in the journal. We also provide a point-by-point response explaining how we have implemented reviewers’ comments. We look forward to the outcome of your assessment.
Comment 1: Authors should present their data as mean plus minus SD and not SEM because readers are interested in knowing the dispersion of value and not the precision of the mean due to paucity of observations at least in one group
Answer: Thank you for your suggestion. We replaced all the data from SE to SD. See page 3, Table 1.
Comment 2: Authors correctly stated as limitations the possible undetected co-morbidities such as T2DM, Metabolic Syndrome and NAFLD and thus they should lessen of importance their findings pointing out with much emphasis this aspect and not using few words.
Answer: We added a paragraph in the Discussion section to describe an altered tryptophan metabolic status in metabolic syndromes. “Many non-communicable diseases including type 2 diabetes, nonalcoholic fatty liver disease and obesity effect to IDO activity in TRP metabolic pathway. Patients with higher TRP level tended to present higher degree of insulin resistance, triglyceride and blood pressure. The IDO was up‐regulated in older patients and related to increase susceptibility of aged liver to nonalcoholic fatty liver disease development”. see page 8 line 206-211.
Round 2
Reviewer 1 Report
The reviewer still believes that when an autopsy is performed, the results are not adding any further information to the determination of the CHD SUD. Indeed the authors confirmed in their responses that the diagnosis and the consequent classification of the two groups was performed looking at the degree of arterial occlusion in heart tissues after death by means of histopathological methods (see answer to comment 4: “The CHD diagnosis was performed by looking at the degree of arterial occlusion and confirmed by pathohistological method”). Furthermore, the authors confirmed the concern of the reviewer that this study describes an altered metabolic status of the individuals before death (see answer to comment 6: “During living individuals, the metabolic status correlated to lesion the disease. We hypothesized that the metabolic status still correlated to the lesion after death. Our results showed that tryptophan and its metabolites in blood correlated with the CHD lesion which the same as living individuals”). In the revised version, the authors stated than in some cases autopsy cannot be performed due to different local confession and culture. Even in these last cases, the results of the manuscript do not add any valuable forensic information, due to the fact that such a profile could not be used to assess the cause of death being able only to identify the risk of a CHD.
For the above mentioned reasons the reviewer still thinks that the manuscript is not suitable for publication on Metabolites.
Reviewer 2 Report
The authors have addressed all my initial concerns and adequately revised their manuscript. I have no further comments.